# A Clean Process for Obtaining High-Quality Cellulose Acetate from Cigarette Butts

**DOI:** 10.3390/ma13214710

**Published:** 2020-10-22

**Authors:** Anna De Fenzo, Michele Giordano, Lucia Sansone

**Affiliations:** 1Department of Chemical, Materials and Production Engineering (DICMAPI), University of Napoli Federico II, p.zzale Tecchio 80, 80125 Napoli, Italy; adefenzo@unina.it; 2Institute for Polymers, Composites and Biomaterials, National Research Council of Italy (CNR), 80055 Portici, Italy; michele.giordano@cnr.it

**Keywords:** cigarette butts, cellulose acetate, extraction process, recovery

## Abstract

The main purpose of this study is to introduce a modified method for recovering and recycling huge number of cigarette butts (CBs) and convert them into a fashion product. CBs are non-biodegradable waste, generally, discarded improperly into the environment. CBs consist of cellulose acetate, which can be recovered through extraction and purification processes. CBs are the most numerically frequent form of waste in the world. A relevant portion of CBS are discarded improperly into the environment. The principal filter components are plasticized cellulose acetate fibers that have the slowest degradation rate (up to years). In fact, a limiting step is the hydrolysis of cellulose acetate polymer into cellulose and acetic acid, which is extremely slow under ambient conditions and represents a relevant environmental risk. A number of studies have been realized on recycling cigarette butts with encouraging results, and several methods have been studied, including recycling of cigarette butts in asphalt concrete and fired clay bricks, as a carbon source, sound-absorbing material, corrosion inhibitor, biofilm carrier, and many more. In this study, we propose a novel, green, low cost, simple, and efficient extraction method of cellulose acetate fibers (CA) from discarded cigarette butts (DCBs). CBs extraction procedure involves a two-step process consisting of washings in water and some subsequent washings in ethanol. The obtained samples of CA are dried at 60 °C for 60 min in the oven. The quality and properties of cellulose acetate extracted and purified are comparable to the pure cellulose acetate fiber used for cigarette butts. The preliminary results obtained on the recovered CA look promising to the use of this recovery material from cigarette butts to obtain a wide consumption fashion product, such as eyeglass frames.

## 1. Introduction

Cigarette butts are recognized as toxic residue, thanks to containing chemical contaminants and residues produced during combustion reaction; they are frequently improperly disposed, thrown on the ground or on beaches. This way, contaminants of cigarette butts are going to be carried by rain into surface water, in agriculture land and thereby pollute the environment and ecosystems. Generally, cigarettes have a length of 85 or 100 mm, and a diameter of about 8 mm, they are made of a filter, tobacco, additives, and cigarette wrapper. Cigarette Filters (CFs) usually have a length of 20 to 30 mm long, so a typical cigarette has 55 to 80 mm of tobacco. The role of the filter is to reduce the quantity of smoke, tar, and fine particles inhaled during the combustion and to decrease the harshness of the smoke, and prevent the formation of tobacco flakes in the smoker’s mouth. The filter captures and retains toxic substances; they also prevent tobacco from entering a smoker’s mouth and supply a mouthpiece that may not collapse because the cigarette is smoked. Filters are generally composed of a plug of acetate cellulose filter tow; the cellulose acetate esters are white and packed tightly together to make a filter. CFs are made from cellulose acetate fibers (CA), which are arranged as microscopic-sized white fibers massed together by glycerol triacetate, they are poorly degradable, with a Y-shaped cross-section that is not perpendicular to the flow. These fibers are synthetic plastic which is similar to cotton. To attach the plug to the cigarette wrapper is used a vinyl resin emulsion. Moreover, the filter plug is protected by a tipping paper, that has the role to connect the filter to the column of tobacco, and not to attach to the lips of smokers. The tobacco leaves have different colors, tastes, burning properties, aromas, and nicotine content, depending on the kind of tobacco and its growing location. Tobacco leaves contain several alkaloids, including nicotine. Nicotine is a toxic alkaloid that causes addiction in smokers and it is a strong insecticide. According to the US Department of Health and Human Services (USDHHS), nicotine raises blood pressure, affects the central nervous system, and constricts blood vessels in humans. Nicotine is a colorless liquid that is soluble in water and is readily absorbed through the skin in its pure form. Moreover, a hundred types of additives are mixed with tobacco during the manufacturing process. Tobacco additives include flavorings such as cocoa, rum, licorice, sugar, and fruit juices, and humectants that are used to keep tobacco moist. The tar is constituted by the substances and additives found in tobacco. These substances are composed of organic and inorganic chemicals, including some carcinogens. According to the USDHHS, smokers are exposed to a toxic mix of over 7000 chemicals when they inhale cigarette smoke. Generally, the paper used to wrap the tobacco is composed of flax or linen fiber; moreover, to control or accelerate the burning rate various chemical substances are added to the paper, such as salts, ammonium phosphate, and sodium and potassium citrates. Moreover, calcium carbonate is added to the paper to ensure the creation of attractive ash as the cigarette burns. The wrappers’ seams are glued with an adhesive that is a modified starch or natural gum [1,2,3].

Although cellulose acetate is a photodegradable polymer, it is not easily biodegradable. It may persist in the environment for years. As already mentioned, this fibrous material is meant to trap tar and other toxic elements during smoking, but when are left on the environment, they can release these substances and this effect implies an increase of risks for the environment. Moreover, in marine ecosystems cigarette butts (CBs) have a big negative impact because the chemicals they contain pose a risk to the organisms of both freshwater and marine environments [4,5,6,7]. The recycling of CBs is difficult as there do not seem to be any easy mechanisms or procedures to assure an efficient and economic separation of the butts, or appropriate treatment of the entrapped chemicals. For this aim, different research groups are concentrating on the likelihood of using cigarette butts within various applications that can be classified into different categories: (1) recover the cellulosic fibers in form of nanofiber/nanocrystalline cellulose (NCC), or direct blend into paper products; (2) additives for manufacturing other materials such as bricks and steel; (3) devices for energy storage; (4) electronic components; (5) chemical and medical components; and (6) alternative materials for sound absorption. Zhao et al. demonstrated the use of recovered cigarette butts in the metallurgical industry as corrosion inhibitor for steel in acid solution [8]. In fact, they show that cigarette butts extracted from water have the flexibility to inhibit corrosion and this aptitude increases with increasing CBs concentration; while Mohajerani et al. realized different types of clay bricks with several weight percentages (from 0% to 10% *w*/*w*) of CBs by mechanical mixing followed by firing at 1050 °C; the fired samples were finally tested for compressive strength, flexural strength, density, and water absorption [9,10]. Ghosh et al. displayed how pyrolysis treatment produces a material with good conductive properties which can be further used in different electronic fields [11]; while Gomez Escobar et al. demonstrated the high sound absorption performance of this material and the potential use of CBs as a component for insulating solutions, which were observed to be competitive even with sale solutions [12]. Ogundare et al. produced NCC from discarded cigarette filters (DCF) [13]. The DCF were treated through an ethanolic extraction, a bleaching in sodium hypochlorite, followed by an alkaline deacetylation, and converted into NCC by sulfuric acid hydrolysis. The study demonstrated that the produced NCC has such a high quality that it could be applied in the field of catalysis or in the biomaterial area. Teixeira et al. developed a cellulose pulp production process from cigarette butts employing an alkaline pulping [14]. Moreover, Abu–Danso et al. studied a new method to recover of cellulose from CBs in the form of cellulose nanofibers and cellulose nanocrystals for removing diclofenac pharmaceutical from water [15]. The great versatility of cellulose acetate outlines strong attention to the problem of recycling cigarette butts. Furthermore, due to the problems related to the thermo-oxidative degradation of this product, it is essential to determine operating conditions and energy parameters that can determine a longer life of products made thanks to recycled materials [16,17,18,19].

In the present work, cigarette filter extraction by means of water and ethanol is a safe and conservative means to reduce filter waste and the recovered CA can be used for obtaining other products. To evaluate the quality of recovered CA, it is characterized by thermogravimetric analysis in the inert and oxidant ambient and compared to unused CA. Moreover, differential calorimetric analysis has been performed to define the glass transition temperature and the possible crystallinity degree of both the samples and FTIR spectroscopy has been executed to delineate the functional groups after the cleaning process and to determine if any difference were present on the recovered CA respect to the unused specimen. Analyzing onset temperatures and mass loss percentages, respectively, in both the used test ambient were adopted two different modeling methods (Kissinger and the Flynn-WallOzawa (FWO) method) [20]. The Kissinger procedure is proposed to estimate activation energies (Ea) in unused CA compared to recovered CA, in which the relative crystallinity, measured at constant heating rates, can be correlated by the Ozawa model with a temperature-independent exponent. The Ea value obtained for unused CA in different test conditions is similar to Ea values of recovered CA; this means that the recovered CA has as degradation kinetics similar to the one of the unused CA. Finally, the extraction and purification reactions do not affect the CA. We investigated, also, the presence of metals, in particular heavy metals, in recovered samples by means of atomic absorption analysis. Moreover, with the recovered CA we have realized a prototype of eyeglass frames.

## 2. Experimental

### Materials and Methods

Used cigarette filters have been collected by Essequadro eyewear Company S.r.l. (Ariano Irpino, Italy). The used cigarette butts were collected from the ashtrays of the local bar and restaurant area of Essequadro Company in Avellino (Ariano Irpino, Italy) and sent to CNR Laboratories in Portici (Portici, Italy), where the cigarette butts, have been sterilized and treated (recovered CA).

As comparing material, we have chosen the Rizla cellulose acetate filters (Rizla, Riz La Croix, France). Rizla is a French brand that produces rolling papers and other related paraphernalia, among these also the cellulose acetate filters, in which tobacco, or marijuana, or a mixture, is rolled to make handmade joints and cigarettes. Morphology, thermal, and chemical properties of the Rizla untreated acetate cellulose filter (unused CA) have been compared to the ones of the recovered CA by means of our extracted methods. Our CA extraction method is based on the washing off the discarded cigarette butts (DCBs) in hot water (50 °C) for 60 min, after external paper removal, CBs have been washed in cold water three times, so to extend CA fiber. Successively to remove potential organic compounds the butts have been washed in ethanol 99% *w*/*w* twice. Finally, the obtained samples of CA were dried at 60 °C for 60 min in the oven. 

Both structural/morphological and thermodynamic/functional properties of unused CA and recovered CA have been studied by means of several experimental techniques to assess the possible material changes. The test analyses have been conducted on five samples of untreated and on five samples of recovered CA.

The morphological properties of samples have been analyzed out by means of an optical microscopy analysis, (OLYMPUS BX51 microscope, Olympus, VA, USA) and by means of a scanning electron microscopy (SEM) using a field emission instrument Quanta 200 FEG. Samples were covered with a layer of gold/palladium alloy in a high resolution metalized Emitech K575X. X-ray microanalysis was carried out by EDX Inca Oxford 250 instrument.

The thermodynamic properties have been evaluated by means of a differential scanning calorimetry (DSC) TA Instruments Discovery DSC (TA Instruments, New Castle, DE, USA using a double scan at 10 °C/min from −40 °C to 250 °C) and by means of thermogravimetric analysis (TA Instruments Q500 TGA, TA Instruments, New Castle, DE, USA) under dynamic conditions with temperature ramp of 5, 7.5, 10, 15, and 20 °C/min from ambient to 800 °C in inert and oxidant atmosphere. Sample mass was approximately 10 mg. Collected TGA data have been modelled by Kissinger and Flynn–Ozawa–Wall method.

In fact, for dynamic TGA measurements, mass loss is monitored as function of temperature at different heating rates. Kinetics degradation in its general form can be modelled as follows:(1)$$\frac{d\alpha }{\mathrm{dt}}=f\left(\alpha \left(T\right)\right)$$

The degradation kinetic analysis was done using the Kissinger and Flynn–Wall–Ozawa methods.

The Kissinger method is based on the following Equation
(2)$$\ln\left(\frac{\beta }{T^{2}}\right)-\ln\left(\frac{\mathrm{AR}}{E_{a}\;\times \;g\left(\alpha \right)}\right)-\frac{E_{a}}{\mathrm{RT}}$$
where α = fraction of conversion (defined as mass loss at respective temperature), A = pre-exponential factor, and g(α) = algebraic expression for integral methods [20,21,22].

From the TGA curves recorded at different heating rates β, temperatures T were determined at the conversions a = 10–90%. The activation energies were calculated from the slope of the straight lines ln(β/T^2^) versus 1/T.

The second method is Flynn–Wall–Ozawa (FWO) method [23,24] is a conversional linear method based on the following equation
(3)$$\ln\beta -c-0.4567\frac{E_{a}}{\mathrm{RT}}$$
where β = heating rate in K min^−1^, c is a constant, T = temperature in K, E_a_ = activation energy in kJ/mol, and R = universal gas constant. The plot log β versus 1/T, obtained from TGA curves recorded at several heating rates, should be a straight line. The activation energy can be evaluated from its slope. The approximation on which both methods are based is that each degradation step can be considered as a reactive process of the first order and the first derivative function results constant [20,25,26,27,28]. The Kissinger and Flynn–Wall–Ozawa methods for calculating the activation energy have the advantage that they do not require knowledge of the reaction mechanism.

Moreover, to understand the chemical structure of samples, Fourier-transform infrared spectroscopy (FTIR) have been used (System 2000 FT-IR, Perkin–Elmer, Waltham, MA, USA). In addition, metals in the samples have been analyzed by means of an atomic absorption spectrometer (A PerkinElmer Analyst 700 atomic absorption spectrometer, Norwalk, CT, USA).

## 3. Results and Discussion

### 3.1. Morphological Characterization

In Figure 1a the images of unused and recovered CA samples, acquired by optical microscopy, are reported. It is observed that the volume of the constituent fibers of specimens, is reduced by the cleaning process.

In Figure 1b, the SEM micrographs for unused and recovered CA samples are reported. It is possible to observe that the fibers constituting the cigarettes filters, after the recovering process, appear with an irregular surface.

In Table 1 we reported the compositional analysis of recovered cigarette butts compared to unused cigarette filters. The concentration of titanium is the same for recovered CA and untreated CA. Generally, Titanium gives the white color to cigarette filters, and as already mentioned nanomaterials made from titanium dioxide are used in cigarette filters to significantly reduce the number of harmful chemicals inhaled by smokers. The % *w*/*w* of element reported in Table 1 is a mean value calculated on different samples analyzed. Based on the data reported in Table 1, we investigated the presence or absence of metals and heavy metals in recovered CA. We examined the presence of heavy metals in recovered CA by flame atomic absorption spectrometer. The results recorded the absence of high concentrations of heavy metals and metals (limit of detection of the instrument, lower than 0.01 mg/L), only 0.04 ppm of titanium was detected. Micevska et al. supposed that the toxicity of butt leaches is attributed to metals and heavy metals. The presence of metals and, in particular, heavy metals in cigarettes is imputed to the growth and cultivation of tobacco, soil contamination, use of pesticide and herbicide, cigarette manufacturing process, and the use of brightening agents on the paper. The presence of Pb and Cd is extremely varied in several cigarette brands [29,30,31].

### 3.2. Thermal Degradation Study

The unused and recovered CA were characterized by TGA and DSC analysis. In the Figure 2a DSC thermograms in the first heating scan are reported for the unused and recovered CA; the unused and recovered samples present a water desorption peak in the range 25–100 °C and, the melting temperature is near 225 °C [32]; the unused CA in the temperature range between 100°C and 210 °C shows a broad peak due to the elimination, probably, of an additive or binder of commercial CA cigarette filter. In Figure 2b DSC thermograms in the second heating scan are reported for the unused CA and recovered CA samples; the measured glass transition for unused CA is 193 °C, while for recovered CA is 192 °C.

In Figure 3a TGA thermograms are reported at 10 °C/min scan rate under the nitrogen inert flow and corresponding derivative and second derivative curves for unused CA sample are reported. It is possible to observe that in the temperature range from ambient to 200 °C one degradation step is present with 10% of mass loss correlated. This mass loss is distributed in two different contributes: The first lied to the sample dehydration with around 2% of mass lost and the second one lied to the decomposition of plasticizer compound present in the process of filters realization (8% mass loss). In the temperature range from 250 °C to 450 °C, the degradation step with 80% of mass lost took place; this degradation step concerns the deacetylation process of CA with maximum rate of degradation corresponding to the temperature of 367 °C. The residue of measurements is 9% of initial weight [13]; while in Figure 3b TG curves at 10 °C/min under the inert flow and corresponding derivative and second derivative curves for recovered CA sample are reported. Compared to the previous graph, it is possible to observe that the first degradation step noticed for unused CA, in this case is absent.

The green process carried out on the dirty filters, erase the traces of plasticizers in the cellulose acetate structure. The principal degradation step occurs in the same temperature range (250−450 °C) and the mass lost in this degradative step is about 85%. The residue of the measurement is about 10% of the initial weight. The cellulose acetate unused is burned within 1.83 min and the thermostability of the CA is not affected by the process since the burning time values are the same for both the investigated samples.

Moreover, to confirm our data in Figure 4 we reported, the thermogravimetric curves of unused CA at different heating rates and corresponding DTG curves under the nitrogen flow. Mass is lost with increasing temperature rate yielding earlier degradation onsets. It can be noticed that the degradation step related to the plasticizer decomposition is independent from the heating scan rate. DTG curves display that maximum peak temperatures are moderately dependent upon heating rates; while Figure 5 shows, respectively, the TG (a) and DTG (b) curves related to the recovered sample at different heating rates under the nitrogen inert flow. It was observed that the absence of degradation step of plasticizers is independent from the heating rate and the deacetylation step is somewhat dependent from the scan rate. Moreover, the residue of all the experiments carried out on unused and recovered CA is independent from the applied scan rate value.

Degradation behavior of the unused CA was also tested in oxidant atmosphere (air flow) and TGA and DTG curves are reported in Figure 6. Figure 7a,b displays thermogravimetric analysis for recovered CA sample under oxidant conditions. Two different degradation steps in the whole investigated temperature range are identified. Slight deviation in the temperature onset with heating rate has been observed corresponding to the deacetylation process (200 °C–400 °C). The charring stage from 400 °C to 550 °C results unaffected by heating rate. The mass loss in this degradation step is 18% for all the applied scan rates. The char yield obtained at 700 °C, in nitrogen flow, is slightly dependent on heating rate, while in the oxidant atmosphere, the mass loss is total at 600 °C and it is not dependent on the heating rate. Moreover, from the ambient temperature to the 220 °C, the identified step under nitrogen flow is related to the dehydration process and plasticizer degradation. The total weight loss in this temperature range is approximately 10% *w*/*w* and result independent from the heating rate. Moreover, under these conditions, in the temperature range from 250 °C to 600 °C, two different degradation steps are traceable. The deacetylation process occurs with maximum peak temperatures increasing with the heating rate whereas the second stage (charring stage) is characterized by an almost constant starting temperature of 400 °C with an end decomposition offsets between 500 °C and 550 °C, showing an irregular increasing trend with the scan rate. A mass loss of 16% in this last step, is noticed independently from heating rate.

Kissinger method has been employed for all TGA scans performed on unused CA and recovered CA, in order to evaluate the activation energy of each degradation step in air and nitrogen flow. A comparison between Kissinger ln(β/Tmax^2^) versus 1/T_max_ curves for the degradation stage identified for both the analyzed samples in the inert and oxidant ambient allowed to collect data discussed below. Regarding to the inert environment (see Figure 8), the deacetylation process of the unused CA sample appears to occur with a greater energy value than the recovered sample (67 KJ/mol for unused CA and 61 KJ/mol for recovered CA). It is worth noting that the dehydration and the plasticizer decomposition take place for the unused sample and the energy contribution of endothermic process of dehydration could be recovered with greater effort to complete the deacetylation process.

As expected, the activation energy values, computed from the measurements carried out under air flow (see Figure 9a,b), confirm that the recovery process provides cellulose acetate equivalent perfectly to the never used cellulose acetate from cigarettes filter. In fact, the activation energy needed for deacetylation process, for both the investigated sample is equal to 40 KJ/mol and the charring stage occur with an energetic contribution of 75 KJ/mol indifferently for unused and recovered CA.

The FOW integral analysis, based on Equation (3) has been also carried out to evaluate the dependence of apparent activation energy over the extent of reaction. While the Kissinger method considers only one point of thermal degradation curve the FOW model examines different points, each corresponding to different conversion values, therefore modeling results are considered only for discussion aims. In Figure 10a,b the energy profiles are reported for unused (red trace) and recovered (blue trace) CA, respectively, in nitrogen and air atmosphere. In inert flow the degradation reaction is activated in the conversion range 20–80% the value of Ea is 200 KJ/mol (see Figure 10a). Under the oxidant atmosphere, when a 10% mass is lost, the energy curve rapidly increases as the decomposition gradually takes place for the investigated samples. However, in the conversion range 15–65%, an energy value of 150 KJ/mol was found for both the analyzed samples. For conversion with α > 65%, the activation energy profiles for both samples are characterized by a positive derivative path toward full conversion. Moreover, two different degradation steps are evident, these are related to the deacetylation process and char formation (see Figure 10b). As expected, also by FOW analysis, the cleaning process of cigarette filters allows to obtain a recovered sample with behavior similar to untreated or “unused” filter.

### 3.3. Characterization of Recovered CA by FTIR Spectroscopy

The spectra of unused sample compared to the measurement carried out on recovered sample is reported in Figure 11. In the region between 3700 cm^−1^ and 3000 cm^−1^ and precisely at 3400 cm^−1^, it is possible to observe the extended band assigned to OH stretching derived from adsorbed OH stretching derived from adsorbed water. Symmetric and asymmetric stretching of C-H methyl group are identified corresponding to 2920 cm^−1^ and 2850 cm^−1^ bands; these bands are connected to sharp peak at 1432cm^−1^ due to –CH_2_ bending. The characteristic bands of cellulose acetate have been highlighted and typical carbonyl stretching band of acetate group is very intense and can be easily identified. These carbonyl band of acetyl groups is visible and appears in an isolated region of FTIR spectrum at 1739 cm^−1^. Bending of the CH group (belonging to methyl or hydroxyl groups in the plane) at 1371 cm^−1^ and a 1214 cm^−1^ band linked to the stretching of the CO bond of the acetyl groups are identified. Finally, the asymmetric stretching of the ester group C–O–C at 1100 cm^−1^ and the vibrational modes of the C-O bond in cellulose molecules, that generate a band centered at 1030 cm^−1^, are noticed [33,34].

By the observation of the FTIR spectra, except for the amount of OH groups on the cellulose structure (unused and recovery), it is possible to affirm and confirm that the extraction process produces cellulose acetate compatible with the unused acetate and reusable in different applications.

### 3.4. Reprocessing Waste Cigarette Butts into Usable Material

The best sustainable solution for cigarette butt’s environment problems is to recycle cigarette butts and use them for the manufacturing of some useable products. We have demonstrated a clean extraction method of the plastic cellulose acetate from the waste cigarette butts. We have been able to reduce the toxicity level of the used butts and remove the bad sniff from them; we can certainly use them for usable products manufacturing, as for example to produce fashionable products such as eyeframes.

Before extruding our dried recovered CA, we have realized a cast transparent film; in fact, the CA recovered has been dissolved in acetone (5% *w*/*w*) and cast in a petri dish. We obtained a very transparent film (see Figure 12).

The film in Figure 12 is realized by recovered CA dissolved in acetone. However, the CA film exhibits some defects, the film is brittle so a plasticizer needs to be added to attain toughness. The choice of plasticizer is based on compatibility with CA, so we must improve the mechanical properties and thermal stability without affecting the film transparency.

The recovered CA granules have been extruded to obtain a brittle, white film that is 2 mm thick.

Now we are studying and testing the plasticizers and pigments, commercial and biodegradable, aimed at giving the desired workability, flexibility, toughness, and colors to our recovered material.

## 4. Conclusions

Cigarette waste pollutes the environment, so this the desired problem to be addressed by humans. However, such waste can be recycled by converting it into raw material for the manufacture of new products. Cigarette butt’s composition includes a variety of compounds, including aromatic and heterocyclic amines, carbonylated compounds, phenols, polycyclic aromatic hydrocarbons, carbon and nitrogen oxides, and ammonia. These compounds show different solubility according to their polarity. With this fact, and considering that cellulose acetate is the main component and it is soluble in acetone or ethyl acetate, we propose a methodology to purify this polymer-based on several solid-liquid extraction steps using solvents with different polarity. Fourier-transform infrared spectroscopy and thermogravimetric analysis (TGA) were employed to characterize the structural and thermal properties of purified CA compared to unused CA. Moreover, the degradation thermal model computed by Kissinger and Ozawa methods were used to calculate the Ea value obtained for recovered and unused CA and to prove the possible variations. We showed that recovered CA has the same chemical, thermal properties of unused CA, the purification method does not alter the material properties. The recycling of CBs and turning this waste into a resource can be a solution to cigarette butt pollution, the important result we reached is that we can obtain a transparent film from acetate cellulose recovered by cigarette butts using a facile and green extraction method.

## Figures and Tables

**Figure 1 materials-13-04710-f001:**
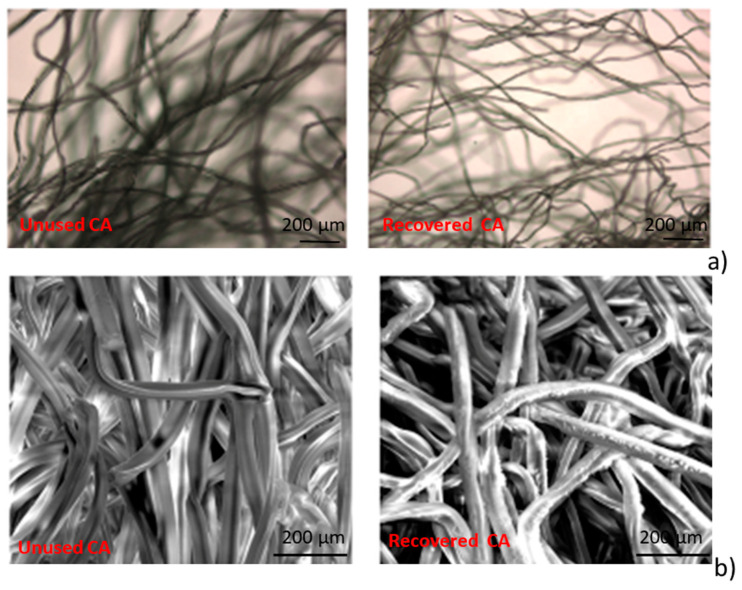
(**a**) Optical microscopy image (50×) of cellulose acetate fibers (CA) unused (left) and CA recovered (right); (**b**) SEM micrographs for CA unused (left) and CA recovered (right) at 800× magnification.

**Figure 2 materials-13-04710-f002:**
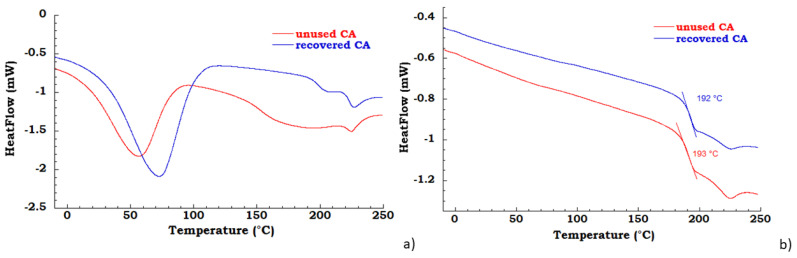
(**a**) differential scanning calorimetry (DSC) thermograms in the first heating scan of unused CA and recovered CA (**b**) DSC thermograms in the second heating scan of unused CA and recovered CA.

**Figure 3 materials-13-04710-f003:**
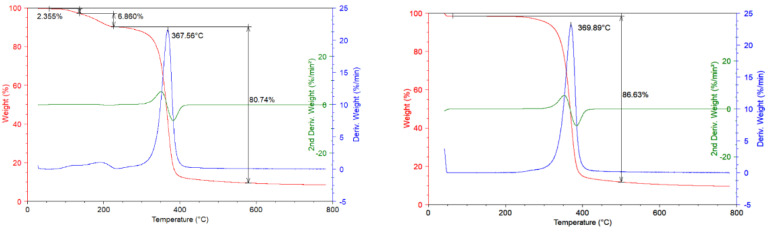
(**a**) TGA Thermograms on unused CA sample (10 °C/min, nitrogen flow); (**b**) TGA Thermograms on recovered CA sample (10 °C/min, nitrogen flow).

**Figure 4 materials-13-04710-f004:**
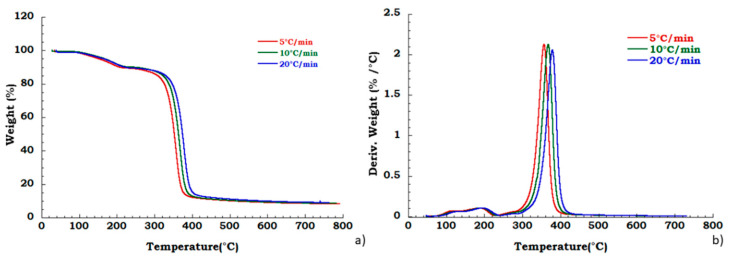
Thermogravimetric curves in nitrogen flow (**a**) unused CA at different heating rate; (**b**) DTG curves for unused CA.

**Figure 5 materials-13-04710-f005:**
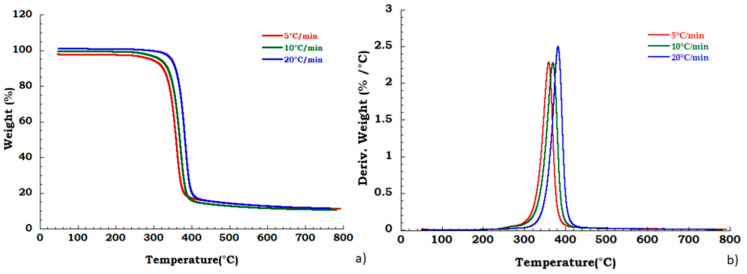
Thermogravimetric curves in nitrogen flow (**a**) TGA curves for recovered CA at different heating rate; (**b**) DTG curves for recovered CA at different heating rate.

**Figure 6 materials-13-04710-f006:**
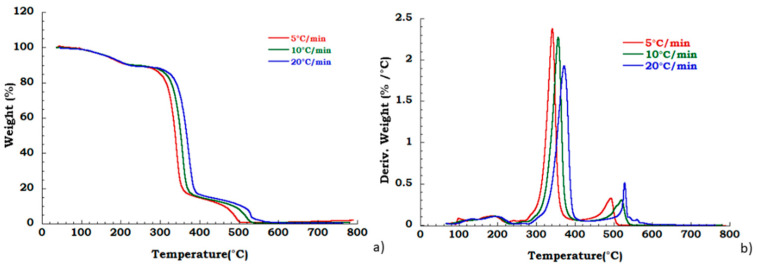
Thermogravimetric curves in air flow (**a**) unused CA at different heating rate; (**b**) DTG curves for unused CA.

**Figure 7 materials-13-04710-f007:**
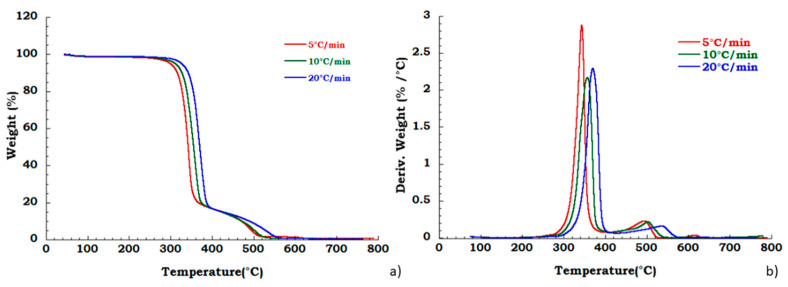
Thermogravimetric curves in air flow (**a**) TGA curves for recovered CA at different heating rate; (**b**) DTG curves for recovered CA at different heating rate.

**Figure 8 materials-13-04710-f008:**
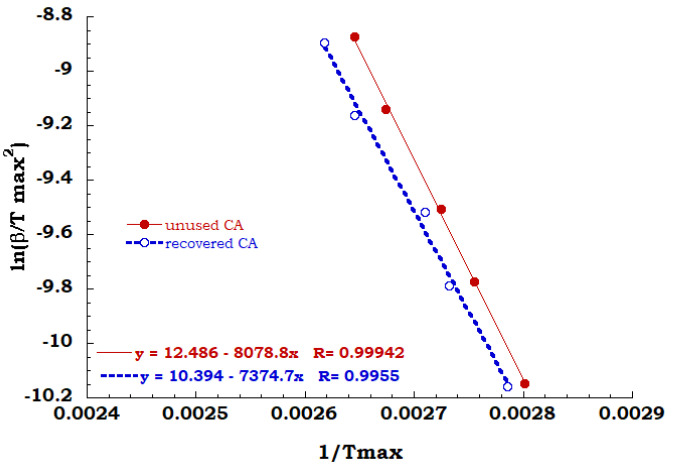
Comparison between Kissinger data obtained for the principal degradation stage.

**Figure 9 materials-13-04710-f009:**
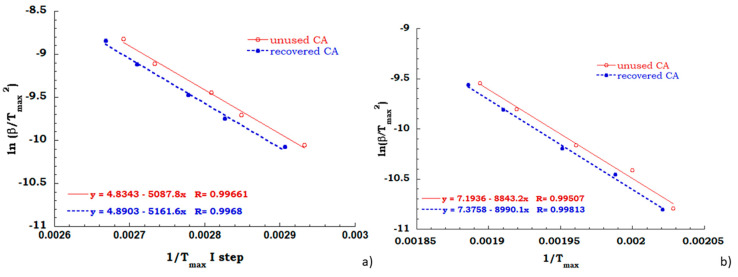
Kissinger data obtained for the deacetylation step (**a**) and charring stage (**b**) in air flow.

**Figure 10 materials-13-04710-f010:**
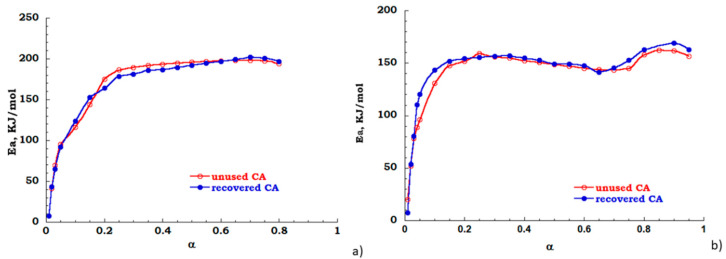
Energy profile derived from FOW (**a**) nitrogen flow; (**b**) air flow.

**Figure 11 materials-13-04710-f011:**
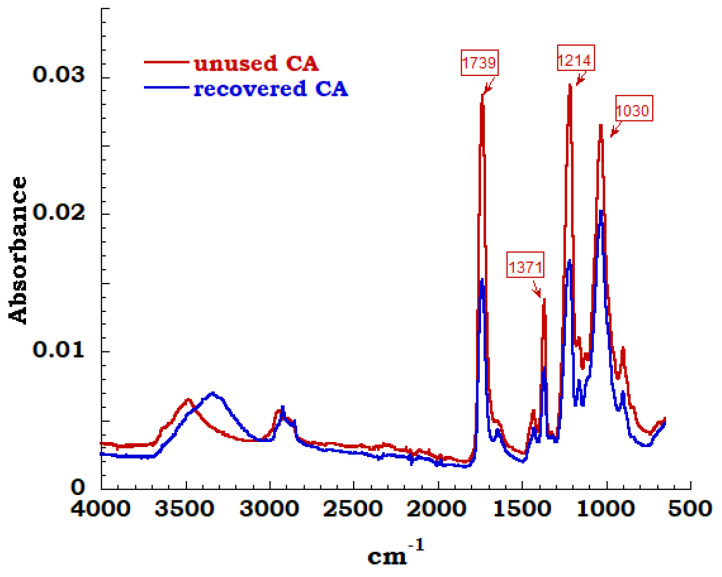
FTIR spectrum on CA unused and recovered.

**Figure 12 materials-13-04710-f012:**
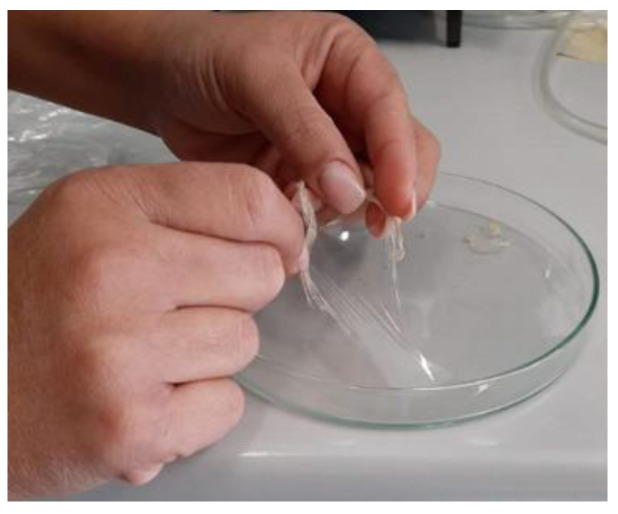
Transparent film of cellulose acetate recovered from cigarette butts.

**Table 1 materials-13-04710-t001:** Chemical composition related to the EDX microanalysis.

Sample	Element	% *w*/*w*
CA unused	C	49.3
	O	50.5
	Ti	0.2
CA recovered	C	50.7
	O	49.1
	Ti	0.2

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
