# Peer review of "A Clean Process for Obtaining High-Quality Cellulose Acetate from Cigarette Butts"

_materials, 2020, doi:10.3390/ma13214710_

Round 1

Reviewer 1 Report

Dear Authors,

your work is very actual and important from recycling and reducing waste from the beaches, for example, where we can find so much cigarette butts as waste.
You are used instrumental techniqes for detection original (unused) and recovered resource celullose acetate which is well known material for product the cigarettes butts.
All results are explained well, but this is preliminary results and a lot of work you need to do, to obtain optimal and reproducible results for production new products, like eyeglass frames as fashion accessories, with clean process. But the topic is so actual!

Because of that please, check the whole paper and make technical changes according to instructions for authors:
1. when you have chapters with indented text, no space between sections is required,
2. edit the text and titles, through the paper, font of text, equations,...
3. edit the figures - must be clear, of the same dimensions and well defined as table or figure.
4. check the language style and same of sentences are not so clear.
5. when you dried sample at for example at 50 oC or higher temperature for 60 or 120 minutes (avoid the abbrev. - h)
6. Check whole paper and uniform my suggestions.

The paper need technical and language edition. It is confusing, hard for reading, it must be redesigned for readers.

With best regards,

Reviewer

Author Response

Dear Authors,

your work is very actual and important from recycling and reducing waste from the beaches, for example, where we can find so much cigarette butts as waste. You are used instrumental techniqes for detection original (unused) and recovered resource celullose acetate which is well known material for product the cigarettes butts. All results are explained well, but this is preliminary results and a lot of work you need to do, to obtain optimal and reproducible results for production new products, like eyeglass frames as fashion accessories, with clean process. But the topic is so actual!

We thank the reviewer for the positive feedback and suggestions.

Because of that please, check the whole paper and make technical changes according to instructions for authors:

  1. when you have chapters with indented text, no space between sections is required

We thank the reviewer for his suggestion, based on this comment we have revised the text manuscript.

  1. edit the text and titles, through the paper, font of text, equations

We thank the reviewer for his suggestion, based on this comment we have changed the text, titles, equations in the manuscript.

  1. edit the figures - must be clear, of the same dimensions and well defined as table or figure.

We thank the reviewer for his observation, based on this, we structured figures and tables in the manuscript

  1. check the language style and same of sentences are not so clear.

We thank the reviewer for his suggestion, based on these we have checked the language style and clarified some sentences in the manuscript.

  1. when you dried sample at for example at 50oC or higher temperature for 60 or 120 minutes (avoid the abbrev. - h)

We thank the reviewer for suggestion, so in the text of the manuscript we changed the abbrev. h as suggested in minutes.

  1. Check whole paper and uniform my suggestions.

We thank the reviewer for observations and suggestions and we have checked the whole paper

The paper need technical and language edition. It is confusing, hard for reading, it must be redesigned for readers.

With best regards,

Reviewer 2 Report

First of all, the quality of English written language should be substantially improved. Otherwise, I have quite some remarks. See the attachment. One of more important remarks is that there is a lack of statistical data; it is not clear how many samples / number of measurements were used / applied for the described analytical / characterisation methods. 

Author Response

# Reviewer 2

General comment: I suggest the authors improve the quality of English written language. Just to illustrate, because of the low quality of English written language, the 2nd sentence in the 2nd paragraph on the page 2 of pdf, starting with “They realize different quite of clay bricks…” is really hardly understandable. This remark is valid for the whole 2nd paragraph on page 2 and I believe in many other cases in the submitted paper

  • Page 1, 3rd paragraph: the reference “Ogundare et al.” – is it “Ogundare” or “Ogundarea” as written in the list of references?

We thank the reviewer for his observation, the reference is ‘’Ogundare et al’’., we have corrected in the Reference List.

  • Page 1, the last paragraph: the reference “Ghosh et al.” – is it “Ghosh” or “Ghosha” as written in the list of references?

We thank the reviewer for his observation, the reference is ‘’Ghosh et al.’’, we have corrected in the Reference List

  • Page 1, 3rd paragraph: what does it mean “thermos-oxidative degradation”?

We thank the reviewer for his observation, this is an error, the correct form is thermo-oxidative degradation of the thermal-oxidative degradation of the material. We have corrected it in the text in the manuscript.

  • Page 3, 4th paragraph: please specify – spell out - what does it mean “Ea value” in the last line of the 4th paragraph (“comparable to the Ea value obtained for unused CA in both the test conditions”).

We thank the reviewer for his suggestion, based on this we have modified the text in the manuscript:

“The Kissinger procedure is proposed to estimate activation energies (Ea) in unused CA compared to recovered CA, in which the relative crystallinity, measured at constant heating rates, can be correlated by the Ozawa model with a temperature-independent exponent. The Ea value obtained for unused CA in different test conditions is similar to Ea values of recovered CA; this means that the recovered CA has a s degradation kinetic similar to the one of the unused CA. Finally, the extraction and purification reaction do not affect the CA.”

  • Page 4, 1st sentence: It would be informative if you could specify where the filters where collected (on the street, on the beach, somewhere else) and were they collected without any selection (so the recent ones as well as very old, washed ones), or preference was given to a certain type of the butts (for instance just very fresh ones). This might be important for interpretation of the results obtained by your study.

We thank the reviewer for his observation, the cigarette butts used were collected from the ashtrays of the local bar and restaurant area of Essequadro Company in Avellino (Italy), there are not preferences of a certain type of butts, neither we have selected them nor we have chosen them; so based on this comment, we have specified in the text manuscript.

‘’Used cigarette filters have been collected by Essequadro eyewear Company S.r.l. (Italy). The used cigarette butts were collected from the ashtrays of the local bar and restaurant area of Essequadro Company in Avellino (Italy) and sent to CNR Laboratories in Portici (Italy), where the cigarette butts, have been sterilized and treated”

  • Page 4, 2nd sentence: Due to low quality of English written language, the 2nd sentence is not easily understandable. What did you mean by “easily retrieval”?

We thank the reviewer for his question, the sentence: ‘’The raw cellulose acetate is commercially available and easily retrieval’’ we want to justify the choice to analyze of cellulose acetate from RIZLA filter. Based on this question we have modified the text in the manuscript:

‘’ As comparing material, we have chosen the Rizla cellulose acetate filters. Rizla is a French brand that produces rolling papers and other related paraphernalia, among these also the cellulose acetate filters, in which tobacco, or marijuana, or a mixture, is rolled to make handmade joints and cigarettes. Morphology, thermal and chemical properties of the Rizla untreated acetate cellulose filter have been compared to the ones of the recovered CA by means of our extracted methods.’’

  • Page 4: please provide references for the described Kissinger and Flyn-Wall-Ozawa methods!

We thank the reviewer for his suggestion; so based on this we have added references in the Reference List

H.E. Kissinger Reaction Kinetics in Differential Thermal Analysis. Anal. Chem., 1957, 29, 1702–1706.

ASTM. ASTM E698-05 Standard Test Method for Arrhenius Kinetic Constants for Thermally Unstable Materials; ASTM International: West Conshohocken, PA, USA, 2005.

  1. H. Flynn, A General Differential Technique for the Determination of Parameters for d(α)/dt = f(α)A exp (−E/RT) Energy of Activation, Preexponential Factor and Order of Reaction (when Applicable), J. Therm. Anal., 1991, 37, 293−305.
  2. H. Flynn, The Isoconversional Method for Determination of Energy of Activation at Constant Heating rates. Corrections for the Doyle Approximation., J. Therm. Anal., 1983, 27, 95-102.
  • Page 4: It is written “The approximation on which both methods are based is that each degradation step can be considered as a reactive process of the first order and the first derivative function results constant.” Why can be considered so? Please provide justification (maybe supported by relevant reference(s)) for this approximation!

We thank the reviewer for his question. A thermogravimetric analyzer was employed to investigate the thermal behavior and extract the kinetic parameters of cellulose acetate. There are different techniques for analyzing the kinetics of solid-state reactions that can generally be classified into two categories: model-fitting and model-free methods. Historically, model-fitting methods are broadly used in solid-state kinetics and show an excellent fit to the experimental data but produce uncertain kinetic parameters especially for nonisothermal conditions. In this work, different model-free techniques such as the Kissinger method and the isoconversional methods of Ozawa, are employed order to analyze nonisothermal kinetic data and investigate the thermal behavior of cellulose acetate. There are a number of approaches for modelling the complex process. The simplest is the empirical model, which employs global kinetics, where the Arrhenius expression is used to correlate the rates of mass loss with temperature. In the calculation of the kinetic parameters the equation conversion function ?(?) for the solid-state reactions is dipendent from the reaction order and many autors have considered of the first order and constant. We have added references in the text manuscript.

  1. D. Doyle, Kinetic Analysis of Thermogravimetric Data, Journal of Applied Polymer Science, 1961, 15, 285-292
  2. Sbirrazzuoli, L. Vincent, A. Mija, and N. Guigo, Integral, differential and advanced isoconversional methods: complex mechanisms and isothermal predicted conversion-time curves, Chemometrics and Intelligent Laboratory Systems, 2009, 96 (2), 219–226.
  3. Zsako, Kinetic analysis of thermogravimetric data, The Journal of Physical Chemistry, 1968, 72 (7), 2406–2411.
  4. Miura, New and simple method to estimate f(E) and k0(E) in the distributed activation energy model from three sets of experimental data, Energy and Fuels,1995, 9 (2), 302–307.
  5. O. Aboyade, M. Carrier, E. L. Meyer, J. H. Knoetze, and J. F. Gorgens, Model fitting kinetic analysis and characterisation of the devolatilization of coal blends with corn and sugarcane residues, Thermochimica Acta, 2012, 530, 95–106.
  • Pages 4 and 5: when describing analytical / characterisation methods, please provide some data about the number of measurements / number of samples, etc, so that the one who reads the paper could get an approximate impression about statistical value(s) of your results

We thank the reviewer for his suggestion; so based on this we have added in the paper that the analyses have been conducted on five samples of untreated and recovered CA:

‘’The test analyses have been conducted on five samples of untreated and on five samples of recovered CA’’

  • Figure 1: The table inserted in Figure 1, containing data about chemical composition is too small.

We thank the reviewer for his observation, based on this we have separated table from Figure 1

  • Page 6, 1st paragraph: “Moreover, by EDX analysis it was possible to derive that the fibers are covered with a titanium layer that partly deteriorates after the process.” Where do you see this from? How do you know this?

We thank the reviewer for his questions. In fact, we have observed and analyzed some different samples and in a different areas of the same sample by SEM-EDX analysis. The data reported in Table 1 are the mean value calculated on different samples. So in some SEM image, the titanium concentration was high, in another low. Based on these questions we have added in text manuscript:

‘’In Table 1 we reported the compositional analysis of recovered cigarette butts compared to unused cigarette filters. The concentration of Titanium is the same for recovered CA and untreated CA. Generally, Titanium gives the white color to cigarette filters, and as already mentioned nanomaterials made from titanium dioxide are used in cigarette filters to significantly reduce the number of harmful chemicals inhaled by smokers. The % w/w of element reported in Table 1 is a mean value calculated on different samples analyzed. Based on the data reported in Table 1, we investigated the presence or absence the metals and heavy metals in recovered CA. We examined the presence of heavy metals in recovered CA by flame atomic absorption spectrometer. The results recorded the absence of high concentrations of heavy metals and metals (limit of detection of instrument, lower than 0.01 mg/L) only 0.04 ppm of titanium was detected. Micevska et al. believe that the toxicity of butt leaches is also due to heavy and trace metals. The presence of metals and, in particular, heavy metals in cigarettes will be attributed to the growth and cultivation of tobacco, soil contamination, pesticide and herbicide application, cigarette manufacturing process, and therefore the use of brightening agents on the paper. The presence of Pb and Cd is extremely varied in several cigarette brands’’

  • Page 6, 1st paragraph: “Moreover, these data have been confirmed by the research of the heavy metals by flame atomic absorption spectrometer. The results recorded the absence of high concentrations of heavy metals and metals, and also the presence of 0.04 ppm of titanium” Where are these data? Is the mentioned absence of high concentrations of heavy metals related to unused or to used butts or to both? And what is meant for you “high concentration of heavy metal”?

We thank the reviewer for his questions. From table 1 EDX analysis we expected for founding concentration of heavy metals in samples, in particular on recovered CA. So we investigated the presence of metals and heavy metals in recovered CA by flame atomic absorption spectrometer. The results recorded show the absence of concentrations of heavy metals and metals in recovered cigarette butts (limit of detection of the instrument, lower than 0.01 mg/L) only 0.04 ppm of titanium was detected, confirming the EDX data. With ‘’high concentration’’ we expected to found high levels of metals and heavy metals that cause environmental pollution and human health risk (for example Pb superior to 0.01 mg/L or Cd superior to 0.003 mg/L, law parameters for estimating human health risk).

Based on these questions, we have changed in the manuscript:

‘’Based on the data reported in Table 1, we investigated the presence or absence of the metals and heavy metals in recovered CA. We examined the presence of heavy metals in recovered CA by flame atomic absorption spectrometer. The results recorded the absence of high concentrations of heavy metals and metals (limit of detection of the instrument, lower than 0.01 mg/L) only 0.04 ppm of titanium was detected. Micevska et al. believe that the toxicity of butt leaches is also due to heavy and trace metals. The presence of metals and, in particular, heavy metals in cigarettes will be attributed to the growth and cultivation of tobacco, soil contamination, pesticide and herbicide application, cigarette manufacturing process, and therefore the use of brightening agents on the paper. The presence of Pb and Cd is extremely varied in several cigarette brands 29-31’’

  • Page 6, 1st paragraph: When mentioning references 18-20 it would be very interesting and informative if you could compare concentrations of heavy metals obtained by you with those in the cited literature.

We thank the reviewer for his suggestion; Moerman and Potts (2011) (Moerman, J., Potts, G., 2011. Analysis of metals leached from smoked cigarette litter compared to unsmoke cigarette filter. Tobacco Control 20, 30–35.) investigated the leaching behavior of heavy metals in CBs. The main purpose of the study was to determine the concentration of metals leached and the effect on the surrounding environment. It was found the metals studied have different leaching behaviors over time. A direct relationship was found between the concentration of leached metals and the soaking period. An increase in metal concentration leached for Ba, Fe, Mn, and Sr was seen over time, while no significant change in metal concentration was seen for Ti, Pb, Zn, and Ni. In contrast, a decrease in metal concentration for Cu, Al, Cr, and Cd was observed over time. This indicates that a piece of cigarette litter is a point source of metal contamination for a least a month. Moreover, it suggests that the longer the waste remains in the environment, the greater the contamination of Ba, Fe, Mn, and Sr. We have studied and measured by means of atomic absorption spectrometer the metals Ba, Fe, Mn, and Sn in recovered CA, but we have not measured metals in CBs. We found titanium at a concentration of 0.04 mg/L (0.04 µg/g), the other metals showed lower values (lower of instrument limit of detection) Moerman et al. reported for Ti a concentration of 0.7 µg/g for CBs, and 0.7 µg/g for unsmoked cigarette filter measured by atomic absorption spectrometer. So the concentration of Titanium in recovered CA is lower compared to literature papers, although it would be very interesting to study metals on untreated CBs to understand if our green process has an effect on the metals.

  • Figure 2, the title: What does “I” in “DSC measurements I scan different temperature range” mean?

We thank the reviewer for his question. Calorimetry is a primary technique for measuring the thermal properties of materials to establish a connection between temperature and specific physical properties of substances and is the only method for direct determination of the enthalpy associated with the process of interest. Differential scanning calorimeter (DSC) is a thermal analysis apparatus measuring how physical properties of a sample change, along with temperature against time. Measurements have been conducted under nitrogen or air atmosphere. Samples have been first heated from -40°C to 250°C and maintained there for 2 min (first scan, I scan), then subsequently cooled down to −40 °C, followed by the second heating to 250 °C (second scan, II scan). Both the heating and cooling rate were 10 °C/min. Based on the observation of reviewer we have changed the Figure 2 caption:

‘’Figure 2 a) and b) DSC thermograms in the first heating scan of unused CA and recovered CA’’

  • Page 6, the last paragraph: “This hypothesis has confirmed from a subsequent scan in heating at the end of the first test where it is possible to observe the absence of this peak”. Where can be seen this?

We thank the reviewer for his question. We have changed in the manuscript Figure 2 and we have added comments:

‘’In the Figure 2 a) DSC thermograms in the first heating scan are reported for the unused CA sample and recovered CA; the unused and recovered samples present a water desorption peak in the range 25-100 °C and, the melting temperature is near 225 °C32; the unused CA in the temperature range between 100°C and 210°C shows a broad peak due to the elimination, probably, of an additive or binder of commercial CA cigarette filter. In Figure 2 b) DSC thermograms in the second heating scan are reported for the unused CA and recovered CA samples; the measured glass transition for unused CA is 193°, while for recovered CA is 192 °C.’’

  • Pages 10&11, FTIR: Using the term “band” instead of “peak” when describing FTIR spectra is preferred. And the word string “lied to OH…« is strange. I would say »…it is possible to observe the extended band assigned to OH stretching derived from adsorbed…«. Please apply elsewhere in the paper!

We thank the reviewer for his suggestions. Based on these we have changed in the manuscript.

‘’The spectra of CA unused sample compared to the measurement carried out on CA recovered sample is reported in Figure 11. In the region between 3700 cm-1 and 3000 cm-1and precisely at 3400cm-1, it is possible to observe the extended band assigned to OH stretching derived from adsorbed OH stretching derived from adsorbed water. Symmetric and asymmetric stretching of C-H methyl group are identified corresponding to 2920 cm-1 and 2850 cm-1 bands; these bands are connected to sharp peak at 1432cm-1 due to –CH2 bending. The characteristic bands of cellulose acetate have been highlighted and typical carbonyl stretching band of acetate group is very intense and can be easily identified. These carbonyl band of acetyl groups is visible and appears in an isolated region of FTIR spectrum at 1739 cm−1. Bending of the CH group (belonging to methyl or hydroxyl groups in the plane) at 1371 cm-1and a 1214 cm-1 band linked to the stretching of the CO bond of the acetyl groups are identified. Finally, the asymmetric stretching of the ester group C-O-C at 1100 cm-1 and the vibrational modes of the C-O bond in cellulose molecules, that generate a band centered at 1030 cm-1, are noticed33, 34 .

  • Pages 10&11, FTIR: Quite some bands and assignments to vibrations of bonds in various functional groups are mentioned. I suggest that you add literature references where these bands and their assignments are mentioned.

We thank the reviewer for his suggestions. We have added references.

  1. Fei, L. Liao, B. Cheng and J. Song, Quantitative analysis of cellulose acetate with a high degree of substitution by FTIR and its application, Anal. Methods, 2017, 9, 6194-6201.
  2. R. Filho, D. S. Monteiro, C. da Silva Meireles, R. M. Nascimentode Assunção, D. A. Cerqueira, H. SilvaBarud, S. J.L. Ribeiro, Y. Messadeq, Synthesis and characterization of cellulose acetate produced from recycled newspaper, Carbohydrate Polymers, 2008, 73 (1), 74-82.
  • Conclusions: “Preliminary tests by differential scanning calorimetry were used to verify the presence of crystallinity areas in the filters structure.” Really? Crystallinity areas are mentioned only in the Introduction (Page 3) and here in the Conclusions. And nowhere else!? By my opinion it is not appropriate to mention something in the Conclusions when it is not at all mentioned in the paper (in the Results & Discussion Section).

We thank the reviewer for his observation. We agree with the reviewer, also because in order to be able to talk about the crystallinity of the material other investigations must be addressed (WAXS Wide Angle X-ray scattering or XRD X ray diffraction). So we have modified Conclusion in the manuscript.

Reviewer 3 Report

This paper presents a easy water/EtOH washing method to treat the recycled cigarette butts, which can be used to cast transparent CA films. The aim is meaningful. Detailed characterizations are carried out about the thermal degradation. However, the introduction is not written clearly with too many irrelevant details. Different parts in discussion are in need of connections. More importantly, the characterizations of the final CA films are missing. Without this final material demonstration, the feasibility of the present recycling process cannot be judged. Thus major revision is recommended.

Comments:

1. It is strange not to write the abstract in a single paragraph. Please modify this.

2. Intro Part – is cellulose acetate fibers the only component in cigarette? Any others? It works as a filter, so what particles/chemicals can it filter so that those things will not go into peoples’ lung?

3. “Applications of cigarette butts in the acoustic field has been hypothesized by Gomez Escobar et al.” “Ogundare et al., produced nanocrystalline cellulose (NCC) produced from discarded cigarette filters (DCF).” Put REF after the first sentence instead of at the end of each paragraph.

4. Intro Part – Each paragraph describes one work, which contains too much detailed information. It just dilutes the whole message. Please try to shorten it. Maybe try to divide those works in different categories: 1) recover the cellulosic fibers in form of nanofiber/NCC, or direct blend into paper products; 2) additives for manufacturing other materials such as bricks and steel; 3) acoustic materials (what form of the products? Film?)

5. Intro Part - “We investigated, also the presence of metals, in particular heavy metals, in recovered samples by means of atomic absorption analysis.” “Moreover, we have realized with the recovered CA from cigarette butts a prototype of eyeglass frames.” Merge into one paragraph.

6. Materials Part – This is a critical one: what quality of the cigarette filters does the company supply in the manuscript? The burned one has been sorted out

7. Figure 1 - Still not clear why both unused and recovered CA have Ti? If Ti is from tabacco, why it appears in the unused one as well?

8. Figure 6 - Why the peak intensity so different between unused and recovered CA?

9. Figure 7 - The film does not look great in the picture – 2mm in thickness but still not easy to be peeled off? So no standalone films? Importantly, should include more information about the final CA film, since that is the solid results showing the success of the current manuscript.

10. Figure 3, 4, 5 – How are these results helping the final materials (transparent film). In other words, why study these the thermal degradation? Needs more information to link the whole manuscript together.

Author Response

# Reviewer 3

This paper presents a easy water/EtOH washing method to treat the recycled cigarette butts, which can be used to cast transparent CA films. The aim is meaningful. Detailed characterizations are carried out about the thermal degradation. However, the introduction is not written clearly with too many irrelevant details. Different parts in discussion are in need of connections. More importantly, the characterizations of the final CA films are missing. Without this final material demonstration, the feasibility of the present recycling process cannot be judged. Thus major revision is recommended.

Comments:

  1. It is strange not to write the abstract in a single paragraph. Please modify this.

We thank the reviewer for his observation, we have modified the Abstract

  1. Intro Part – is cellulose acetate fibers the only component in cigarette? Any others? It works as a filter, so what particles/chemicals can it filter so that those things will not go into peoples’ lung?

We thank the reviewer for suggestions. Based on this comment we have strongly revised the Intro text in the following way:

‘’Cigarettes are typically 85 or 100 mm long, and have diameters of about 8 mm, they are made of a filter, tobacco, additives and cigarette wrapper. Filters are usually 20 to 30 mm long, so a typical cigarette has 55 to 80 mm of tobacco. A filter is intended to cut back the quantity of smoke, tar and fine particles inhaled during the combustion and to cut back the harshness of the smoke and keep tobacco flakes out of the smoker's mouth. The filter captures and retains toxic substances. they also prevent tobacco from entering a smoker's mouth and supply a mouthpiece that may not collapse because the cigarette is smoked. Filters are generally composed by a plug of acetate cellulose filter tow; the cellulose acetate esters are white, and packed tightly together to make a filter. Cigarette filters are manufactured from about 95% poorly degradable microscopic-sized white fibres massed together, characterized by a Y-shaped cross-section that is not perpendicular to the flow and it is made from CA. These fibres are synthetic plastic which sounds like cotton. In the CFs, the CA fibres are linked to every other through the glycerol triacetate, which could be a plasticizer. CA is one among the foremost important esters of cellulose. A vinyl resin emulsion is employed because the glue to connect the plug to the wrapper and to seam the wrapper. Moreover, a tipping paper, often printed to appear like cork, covers the filter plug and attaches the filter to the column of tobacco, and it has not to adhere to the lips of smokers. The leaves of the tobacco plant have different tastes, burning properties, aromas, color, and nicotine content, depending on the kind of tobacco and its growing location. Tobacco leaves contain several alkaloids, including nicotine. Nicotine is a toxic alkaloid that causes addiction in smokers and it is a strong insecticide. According to the US Department of Health and Human Services (USDHHS), it raises blood pressure, affects the central nervous system and constricts blood vessels in humans. Nicotine is a colorless liquid that is soluble in water, and is readily absorbed through the skin in its pure form. Moreover, hundred types of additives are mixed with tobacco during the manufacturing process. Tobacco additives include flavorings such as cocoa, rum, licorice, sugar, and fruit juices, and humectants that are used to keep tobacco moist. The tar is the substances and additives found in tobacco. The cool Tar seems to be a sticky yellow-brown substance that it is composed of organic and inorganic chemicals, including some carcinogens. According to the USDHHS, smokers are exposed to a toxic mix of over 7000 chemicals when they inhale cigarette smoke. Generally, the paper used to wrap the tobacco is made from flax or linen fiber; to control or accelerate the burning rate various chemical substances are added to the paper, such as salts, ammonium phosphate and sodium and potassium citrates. Moreover, calcium carbonate is added to the paper to ensure the creation of an attractive ash as the cigarette burns. The wrappers' seams are glued with an adhesive that is a modified starch or natural gum 1-3.

  1. “Applications of cigarette butts in the acoustic field has been hypothesized by Gomez Escobar et al.” “Ogundare et al., produced nanocrystalline cellulose (NCC) produced from discarded cigarette filters (DCF).” Put REF after the first sentence instead of at the end of each paragraph.

We thank the reviewer for his comment so based on this comment we have modified in the paper.

  1. Intro Part – Each paragraph describes one work, which contains too much detailed information. It just dilutes the whole message. Please try to shorten it. Maybe try to divide those works in different categories: 1) recover the cellulosic fibers in form of nanofiber/NCC, or direct blend into paper products; 2) additives for manufacturing other materials such as bricks and steel; 3) acoustic materials (what form of the products? Film?)

We thank the reviewer for his observation, based on this we have changed in the manuscript in the following way:

‘’The recycling of CBs is difficult as there do not seem to be any easy mechanisms or procedures to assure an efficient and economic separation of the butts, or appropriate treatment of the entrapped chemicals. For this aim, different research groups are concentrating on the likelihood of using cigarette butts within various applications that can be classified into different categories: 1) recover the cellulosic fibers in form of nanofiber/NCC, or direct blend into paper products; 2) additives for manufacturing other materials such as bricks and steel; 3) devices for energy storage; 4) electronic components; 5) Chemical and medical components; 6) alternative materials for sound absorption’’

  1. Intro Part - “We investigated, also the presence of metals, in particular heavy metals, in recovered samples by means of atomic absorption analysis.” “Moreover, we have realized with the recovered CA from cigarette butts a prototype of eyeglass frames.” Merge into one paragraph.

We thank the reviewer for his observation, so we have merged into one paragraph.

  1. Materials Part – This is a critical one: what quality of the cigarette filters does the company supply in the manuscript? The burned one has been sorted out

We thank the reviewer for his observation. Used cigarette filters have been collected by Essequadro eyewear Company S.r.l. (Italy). The used cigarette butts were collected from the ashtrays of the local bar and restaurant area of Essequadro Company in Avellino (Italy) and sent to CNR Laboratories in Portici (Italy), where the cigarette butts, have been sterilized and treated. Based on this observation we have added in the text manuscript:

‘’Used cigarette filters have been collected by Essequadro eyewear Company S.r.l. (Italy). The used cigarette butts were collected from the ashtrays of the local bar and restaurant area of Essequadro Company in Avellino (Italy) and sent to CNR Laboratories in Portici (Italy), where the cigarette butts, have been sterilized and treated’’

  1. Figure 1 - Still not clear why both unused and recovered CA have Ti? If Ti is from tabacco, why it appears in the unused one as well?

We thank the reviewer for observation. We have specified in the paper in the following way:

‘’In Table 1 we reported the compositional analysis of recovered cigarette butts compared to unused cigarette filters. The concentration of Titanium is the same for recovered CA and untreated CA. Generally, Titanium gives the white color to cigarette filters, and as already mentioned nanomaterials made from titanium dioxide are used in cigarette filters to significantly reduce the number of harmful chemicals inhaled by smokers. The % w/w of element reported in Table 1 is a mean value calculated on different samples analyzed.’’

  1. Figure 6 - Why the peak intensity so different between unused and recovered CA?

We thank the reviewer for his question. In FTIR, an increase in the peak intensity usually means an increase in the amount (per unit volume) of the functional group associated with the molecular bond, whereas a shift in peak position usually means the hybridization state or electron distribution in the molecular bond has changed.

  1. Figure 7 - The film does not look great in the picture – 2mm in thickness but still not easy to be peeled off? So no standalone films? Importantly, should include more information about the final CA film, since that is the solid results showing the success of the current manuscript.

We thank the reviewer for his observations. The film in Figure 7 (now Figure 12) is realized by recovered CA dissolve in acetone. However, the CA film exhibits some defects, the film is brittle so a plasticizer needs to be added to attain toughness. The choice of plasticizer is based on compatibility with CA, must improve the mechanical properties and thermal stability without affecting the film transparency.

  1. Figure 3, 4, 5 – How are these results helping the final materials (transparent film). In other words, why study these the thermal degradation? Needs more information to link the whole manuscript together.

We thank the reviewer for his comment. In the paper, we propose an efficient extraction method of CA from DCBs based on a two-step process consisting of washing in water and some subsequent washings in ethanol. The Fourier-transform infrared spectroscopy (FTIR) and thermogravimetric analysis (TGA) were employed to characterize the structural and thermal properties of purified CA compared to unused CA, to verify and demonstrate that there was or there was not chemical, thermal variation in recovered CA. In particular, the degradation thermal model computed by Kissinger and Ozawa methods was used to calculate the Ea value obtained for recovered and unused CA and to prove the possible variations. We showed that recovered CA has the same chemical, thermal properties of unused CA, the purification method does not alter the material properties. Generally, from commercial cellulose acetate is possible to obtain a transparent film, the important result we reached is that we obtain a transparent film from acetate cellulose recovered by cigarette butts using a facile and green extraction method.

Based on this comment we changed in the text manuscript:

’Cigarette waste pollutes the environment, so this the desired problem to be addressed by humans. However, such waste can be recycled by converting it into raw material for the manufacture of new products.  Cigarette butts composition includes a variety of compounds, including aromatic and heterocyclic amines, carbonylated compounds, phenols, polycyclic aromatic hydrocarbons, carbon and nitrogen oxides, and ammonia. These compounds show different solubility according to their polarity. With this fact, and considering that cellulose acetate is the main component and it is soluble in acetone or ethyl acetate, we propose a methodology to purify this polymer-based on several solid-liquid extraction steps using solvents with different polarity. Fourier-transform infrared spectroscopy (FTIR) and thermogravimetric analysis (TGA) were employed to characterize the structural and thermal properties of purified CA compared to unused CA.  Moreover, the degradation thermal model computed by Kissinger and Ozawa methods were used to calculate the Ea value obtained for recovered and unused CA and to prove the possible variations. We showed that recovered CA has the same chemical, thermal properties of unused CA, the purification method does not alter the material properties. The recycling of CBs and turning this waste into a resource can be a solution to cigarette butt pollution, the important result we reached is that we can obtain a transparent film from acetate cellulose recovered by cigarette butts using a facile and green extraction method’’

Round 2

Reviewer 2 Report

Dear authors, I have carefully read your answers to my questions and comments. I am satisfied with your explanations and changes performed in the text. 

Author Response

We thank the Reviewer for his suggestions and observations, in order to improve our manuscript.

Reviewer 3 Report

Great work on the revision. However, there are still issues need to be solved before the acceptance of the manuscript:

1. Please proofread the whole manuscript! It still contains lots of errors like "see Figure 9(Error! Reference source not found.a) and b))". And so many paragraphs contain only one sentence. why not merge them into one paragraph. Highly recommend to use a "English Polishing" service from either MDPI or other publishers/companies....

2. Critical Issue: what is "unused CA"? This is where it first appears in the manuscript "To evaluate the quality of recovered CA, it is characterized by thermogravimetric analysis in the inert and oxidant ambient and compared to unused CA." Is unused CA the unused cigarette butts? Please be more specific and name it in a better way. 

3. Importantly, the feasibility of making films from recovered cigarette CA still hasn't been proved, although authors state such challenges. Is it really true such CA can be processed into films? What is the quality of the CA even for the unused one? Mw? How the readers know this is a good approach to make "useable materials" without solid data in this manuscript. 

Unfortunately, without this last step of important data, the manuscript reads like an unfinished project.... 

Author Response

Great work on the revision. However, there are still issues need to be solved before the acceptance of the manuscript:

  1. Please proofread the whole manuscript! It still contains lots of errors like "see Figure 9". And so many paragraphs contain only one sentence. why not merge them into one paragraph. Highly recommend to use a "English Polishing" service from either MDPI or other publishers/companies....

We thank the reviewer for his observation; based on this comment we have strongly revised the Manuscript.

  1. Critical Issue: what is "unused CA"? This is where it first appears in the manuscript "To evaluate the quality of recovered CA, it is characterized by thermogravimetric analysis in the inert and oxidant ambient and compared to unused CA." Is unused CA the unused cigarette butts? Please be more specific and name it in a better way.

We thank the reviewer for his suggestion, based on this we have modified the text in the manuscript:

‘’As comparing material, we have chosen the Rizla cellulose acetate filters. Rizla is a French brand that produces rolling papers and other related paraphernalia, among these also the cellulose acetate filters, in which tobacco, or marijuana, or a mixture, is rolled to make handmade joints and cigarettes. Morphology, thermal and chemical properties of the Rizla untreated acetate cellulose filter (unused CA) have been compared to the ones of the recovered CA by means of our extracted methods. Our CA extraction method is based on washing of the DCBs in hot water (50°C) for 60 minutes, after external paper removal, CBs have been washed in cold water three times, so to extend CA fiber. Successively to remove potential organic compounds the butts have been washed in ethanol 99% w/w twice. Finally, the obtained samples of CA were dried at 60°C for 60 minutes in the oven’’.

  1. Importantly, the feasibility of making films from recovered cigarette CA still hasn't been proved, although authors state such challenges. Is it really true such CA can be processed into films? What is the quality of the CA even for the unused one? Mw? How the readers know this is a good approach to make "useable materials" without solid data in this manuscript. Unfortunately, without this last step of important data, the manuscript reads like an unfinished project....

We thank the reviewer for his observation. In this paper, the authors present a facile and green extraction method of cellulose acetate fibers from discarded cigarette butts. By means of this method, it is possible to recover and recycle a huge number of cigarette butts using only two reagents; the extracted cellulose acetate from cigarette butts present the same chemical, physical properties of untreated cellulose acetate used to do cigarette filters. There are not metals, heavy metals and the recovered sample is white. We have also demonstrated that it is possible to extrude the extracted cellulose acetate or to obtain a film by means of solvent cast method, but without the plasticizers, the cast film and extruded film are brittle. The cast film is transparent; this means that our extraction method bleaches the cigarette butts. To obtain commercial acetate cellulose sheets to make eyeglass frames, we must use plasticizers and pigments. Now, we are moving on mechanical tests on recovered cellulose acetate and studying a biodegradable plasticizer such as PEGDA, instead of commercial plasticizers used in cellulose acetate sheet for eyeglass frames. We are also testing the commercial acetate cellulose sheets. As we reported in the last sentence of the abstract, the preliminary results obtained on the recovered CA look promising to the use of this recovery material from cigarette butts to obtain a wide consumption fashion product, such as eyeglass frames. We know that more tests and studies must be done to obtain recovered acetate cellulose sheets to be worked to obtain a fashion product or other products, but our first step is the clean extraction process by means of which we have obtained high-quality cellulose acetate from cigarette butts.
